# Study of Changes in Optical and Heat-Conducting Properties of AlN Ceramics under Irradiation with Kr^15+^ and Xe^22+^ Heavy Ions

**DOI:** 10.3390/nano10122375

**Published:** 2020-11-28

**Authors:** Artem L. Kozlovskiy, Maxim V. Zdorovets, Vladimir V. Uglov

**Affiliations:** 1Engineering Profile Laboratory, L.N. Gumilyov Eurasian National University, Nur-Sultan 010008, Kazakhstan; mzdorovets@inp.kz; 2Laboratory of Solid State Physics, The Institute of Nuclear Physics, Almaty 050032, Kazakhstan; 3Department of Intelligent Information Technologies, Ural Federal University, 620075 Yekaterinburg, Russia; 4Department of Solid State Physics, Belarus State University, 220030 Minsk, Belarus; uglov@bsu.by

**Keywords:** radiation damage, optical properties, aluminum nitride, defects, fission fragments, heavy ions

## Abstract

AlN-based ceramics have great prospects for use in the field of structural materials for reactors of the new generation of GenIV, as well as dosimetric and optical devices. Interest in them is due to their unique physical and chemical properties, high resistance to degradation and excellent insulating properties. This work is devoted to the study of changes in the optical and heat-conducting properties of AlN ceramics as a result of irradiation with Kr^15+^ and Xe^22+^ heavy ions with energies close to those of fission fragments of uranium nuclei, and fluences 10^14^–10^15^ ion/cm^2^. During the study, dose relationships of changes in the optical properties of ceramics were established, as well as the effect of the type of ions on the degree of radiation damage and deterioration of optical characteristics. It has been found that an increase in the irradiation dose for Kr^15+^ ions leads to a slight increase in the depth of electron traps, while for samples irradiated with Xe^22+^ ions there is a sharp increase in the depth of occurrence from 5 to 20%, depending on the irradiation dose. For samples irradiated with Xe^22+^ ions, the greatest decrease in thermal conductivity was 19%, while for ceramics irradiated with Kr^15+^ ions, the maximum decrease was not more than 10%. The results show a significant resistance of ceramics to radiation damage by Kr^15+^ ions and negative effects, leading to a decrease in the resistance of optical and conductive properties of ceramics when irradiated with Xe^22+^ ions with doses higher than 5 × 10^14^ ion/cm^2^. Using the X-ray diffraction method, the dependences of structural distortions and changes in dislocation density in the structure of ceramics on the radiation dose were established. It has been determined that the main structural changes are associated with the fragmentation of grains, which result in an increase in the dislocation density, as well as deformation and distortion of the crystal lattice as a result of the formation of complex defects in the structure.

## 1. Introduction

In recent decades, much attention has been paid to the study of radiation damage in ceramic materials based on oxides, carbides and nitrides [1,2,3]. The interest in these classes of materials is primarily related to their wide potential application in the field of nuclear power and reactor building, where these ceramics can be used as structural materials, as a basis for dosimetric or thermal insulation devices, etc. [4,5,6]. From a fundamental point of view, interest in these materials is due to the possibility of obtaining new knowledge in the field of studying the mechanisms of defect formation, tread formation and radiation resistance of materials [7,8,9]. At the moment, among all nitride ceramics, latent tracks from heavy ions were found in Si_3_N_4_ [10,11], while for AlN, the formation of latent tracks was not found, which the authors of [12] determined with a high threshold bias energy for AlN (>34 keV/nm) required to form tracks. In this regard, high resistance to radiation damage, as well as good insulating and heat-conducting properties, make ceramics based on AlN one of the most promising materials in the field of nuclear energy, structural materials for reactor building, dosimetry, etc. [13,14,15].

The optical properties of nitride ceramics deserve special attention when considering nitride ceramics and their practical application [16,17,18]. Interest in them is due to the possibility to use them in the fields of the abovementioned applications. At the same time, high radiation and chemical resistance indicators open up wide prospects for nitride ceramics in these areas, as well as opportunities to work in conditions of increased radiation background or aggressive media.

A fairly large number of scientific works are devoted to the study of the optical properties of aluminum nitride, as well as various nanostructures and ceramics based on it. For example, a scientific group led by Berzina et al. [19,20,21] carried out a number of studies that were aimed at studying the properties of luminescence of AlN and the effect of structural and impurity defects on their change. They were the first to explain the presence of peaks in the absorption spectra and their relationship with defect and vacancy complexes [19,20,21]. Great attention in the study of the optical properties of AlN is paid to the influence of impurities in the structure of ceramics and their further evolution. For example, a series of works by Trinkler et al. is devoted to the study of changes in optical and luminescent properties by modification with UV and gamma radiation [22,23,24,25]. The authors studied in detail the mechanisms of luminescence change in AlN, which are associated with the recombination processes and changes in the concentration of defects in the crystal lattice. However, while the optical properties of AlN ceramics have been studied, there are still many questions regarding the effects of ion bombardment defect formation processes on the properties of ceramics.

The aim of this work is to study the effect of irradiation with heavy ions Kr^15+^ and Xe^22+^ with energies of 140 MeV and 220 MeV and fluences of 10^14^–10^15^ ion/cm^2^ on the optical and heat-conducting properties of aluminum nitride ceramics. This work is part of the research cycle of our scientific group devoted to the study of the physical and chemical, structural, optical and mechanical properties of ceramics exposed to ionizing radiation. Previously, our research group studied in detail the changes in the structural and mechanical properties of AlN ceramics exposed to irradiation of heavy Fe^7+^ ions with energies up to 100 MeV [26,27]. The mechanisms of helium swelling and processes of hydrogenation of the near-surface layer exposed to high-dose irradiation (10^16^–5 × 10^17^ ion/cm^2^) have been studied [28,29,30]. Studies have been carried out to investigate the effect of the formation of impurity inclusions as a result of ion implantation in the optical and structural properties of ceramics irradiated with low-energy C^2+^ ions [31]. During these studies, the mechanisms of defect formation in the near-surface layer were established, as well as the effect of defects on the change in the properties of ceramics. The results obtained have shown a high resistance of these ceramics to radiation exposure, as well as to processes the degradation of the near-surface layer as a result of the accumulation of radiation-stimulated defects, which opens up prospects for using these materials in nuclear power engineering as structural materials. A distinctive feature of this research is a detailed study of the effect of defect formation under irradiation with heavy ions with energies of 140–220 MeV, comparable to fragments of uranium fission on the optical and luminescent properties of ceramics. As is known, in dielectrics, a change in the optical properties is directly related to a change in the electron density, as well as structural and impurity defects. At the same time, in the case of radiation damage arising from irradiation with heavy ions with an energy of more than 1.0 MeV/nucleon, the greatest contribution to the creation of defects is made by elastic collisions of ions with electron shells.

## 2. Experimental Methods

Commercial polycrystalline AlN ceramics were stabilized with aluminum oxide (not more than 4 at%), which are widely used as an insulating material. Samples were purchased from (Karefonte, Shenzhen JRFT, China). According to the manufacturer’s data, the ceramics were obtained by sintering Al_2_O_3_ powders in a nitrogen (N_2_) atmosphere at a temperature of 1600–1800 °C and a pressure of *p* = 760 mmHg. The thickness of the samples was 20 μm, the density was 3.26 g/cm^3^, the structure type was hexagonal, the symmetry type was P63mc(186); according to X-ray diffraction analysis, the grain size was 170–175 nm, and this size is in good agreement with the observed sizes using a scanning electron microscopy. Before irradiation, the samples were subjected to mechanical polishing of the surface in order to obtain a roughness of no more than 5–10 nm. The choice of a sample thickness of 20 μm is due to the path length of ions in the ceramic in order to simulate defects in the entire irradiated volume.

Interest in this class of nitrides is primarily due to their physical, chemical and insulating properties, as well as their high radiation resistance.

Irradiation of ceramics in order to determine the effect of high-energy ions on changes in properties was carried out on a DC-60 accelerator (INP, Nur-Sultan, Kazakstan). Heavy accelerated Kr^15+^ and Xe^22+^ ions with energies of 140 MeV and 220 MeV, respectively, were chosen as ions. The choice of these ions is due to the possibility of simulating the effects of defect formation arising from irradiation with uranium fission fragments. Irradiation doses were 10^14^–10^15^ ion/cm^2^. The choice of these irradiation doses makes it possible to simulate the influence of the overlapping effect of defect regions that arise in the irradiated material during the passage of heavy ions. For the selected radiation doses, the probability of overlap varies from 100 to 1000. The samples were irradiated in a vacuum chamber on a water-cooled target in order to avoid local overheating of the sample under irradiation. According to the literature [10,32], in the case of irradiation with heavy ions with an energy of more than 100 MeV, when a heavy ion passes through the target material, the radius of the defect region, which in some cases is called a latent track, is estimated to be from 3 to 10–20 nm. This region is a defect structure arising from the interaction of an incident ion with the electronic and nuclear subsystem of the target. An increase in the irradiation fluence above 10^13^ ion/cm^2^ leads to overlapping of these regions [10,32].

The thermally stimulated luminescence (TSL) curves of ceramics before and after irradiation were measured in a temperature range from 50 to 550 °C. For the original samples, luminescence was observed after initialization with X-ray radiation (55 kV, 10 mA) for 10 min. For irradiated samples, luminescence was observed after irradiation. The luminescence was recorded using a Hamamatsu R6357 photomultiplier tube (Hamamatsu Photonics K.K., Shimokanzo, Japan) with a sensitivity in the range of 200–900 nm. The heating rate of the samples was 50 °C/min; contact with the heating element was carried out by applying a silver paste to the ceramic surface. The heating rate was controlled through thermocouples glued to the sample surface.

Optical properties such as transmittance, adsorption and bandgap were determined by taking the UV-Vis spectra with a Jena Specord-250 BU analytical spectrophotometer (AnalyticJena, Jena, Germany) with an integrating sphere. The wavelength range was 190–1100 nm, the scanning step was 1 nm and the scanning speed was 0.5 nm/s. The measurement of thermal conductivity was carried out by measuring the absolute longitudinal heat flux, by heating the samples and measuring the temperature difference using thermocouples (Xian Votesen Electronic Technology Co., Ltd., Xian, China).

Determination of the structural characteristics, as well as the assessment of structural distortions and deformations were carried out using the X-ray diffraction method, which was implemented on a D8 Advance Eco powder diffractometer (Bruker, Karlsruhe, Germany). Shooting conditions: Cu-k*λ* = 1.54 Å, 2*θ* = 30–75°, step = 0.02°.

## 3. Results and Discussion

One of the important properties of ceramics based on aluminum nitride, which have a wide potential application as insulating radiation-resistant materials, as well as the basis for dosimetric devices, is the determination of the kinetics of ceramics degradation as a result of exposure to ionizing radiation, as well as the effect of accumulation of defects at high radiation doses. In this case, an important role is played by the so-called effect of overlap of defect regions arising near the trajectories of the ion passage in the material, as well as subsequent changes in the electronic and nuclear subsystems. In the case of heavy ions with energies above 100 MeV, the main contribution (more than 95–97%) to the mechanisms of defect formation is made by the interactions of incident ions with the electron subsystem and the energy losses of ions due to elastic collisions with electron shells. Moreover, in contrast to metals, in which the knocked-out electrons that interact with ions have high mobility and are able to restore most of the changes in the electron density due to compensation and relaxation, for ceramics, such restoration is difficult due to their insulating properties as well as low charge mobility. In this case, a change in the concentration of defects, as well as a change in the electron density as a result of irradiation, will play a significant role, both in the optical properties and in the thermal insulation properties.

Figure 1 shows the TSL curves of ceramics before and after irradiation with heavy ions with different fluences.

For the initial samples, the TSL curve is characterized by the presence of a small peak in the region of 80–110 °C, which corresponds to the release of trapped electrons from the traps, and a large peak in the region of 280–380 °C with a maximum at 320 °C. An increase in the intensity of the TSL curve above a temperature of 500 °C is due to the thermal radiation of the samples during heating.

For irradiated samples at irradiation fluences of 1 × 10^14^ ion/cm^2^, a decrease in intensity and an insignificant shift of the maximum toward higher temperatures are observed. A further increase in the radiation dose leads to an increase in the trends of displacement of the maxima and a decrease in their intensities (see Figure 1c). Additionally, for irradiation doses of 5 × 10^14^–1 × 10^15^ ion/cm^2^, there is a slight increase in the peak intensity in the range of 80–110 °C, which indicates a change in the defect structure in ceramics at high irradiation doses. In turn, the shift of the maximum to the region of high temperatures is also due to an increase in the defect structure of ceramics as a result of irradiation, as well as a change in the depth of occurrence of electron traps and their activation energy in irradiated ceramics. In the case of irradiation with heavy ions, when they pass through the material in small areas along the trajectory of the ions, a large gradient of the electron density arises due to the large energy losses of the ions. In this case, the overlap of these regions leads to an increase in the concentration of defects associated with the creation of electron-hole pairs, regions with local electronic excitation, breaking of chemical and crystal bonds as well as atomic diffusion in the damaged layer. The appearance of defect regions leads to a change in the activation energy and the depth of occurrence of electron traps, which results in an increase in the activation temperature, as well as a decrease in the area under the curve (light sum), which is associated with a change in the band gap of deep electron traps. Additionally, in the case of large energy losses of incident ions and, consequently, large energy transfers to the electronic and nuclear subsystems, it can lead to the formation of not only cascades of elastic collisions, but also the formation of amorphous regions or regions of disorder, which also have a negative effect on optical and luminescent properties of ceramics.

Figure 2 shows the dependences of the change in the TSL curves of the studied ceramics before and after irradiation with heavy ions. For the initial samples, the maximum of the luminescence peak is observed at a wavelength of 375–380 nm. For samples irradiated with Kr^15+^ ions, an increase in the irradiation dose first leads to a decrease in the intensity of the maximum, as well as to a shift to the region of long waves, which is caused by a change in the defect structure in ceramics. In the case of ceramics irradiated with Kr^15+^ ions, the decrease in intensities, as well as their displacement, is manifested significantly less than for samples irradiated with Xe^22+^ ions. Such a difference in the change in the TSL curves are associated with different energy losses of ions, as well as large energy losses on the electron shells of Xe^22+^ ions, which lead to large changes in the electron density in the irradiated material. Figure 2c shows a graph of the dependence of the change in the intensity of TSL curves on the radiation dose for various types of ions. As can be seen from this graph, in the case of irradiation with Xe^22+^ ions, the decrease in intensity is more pronounced than in the case of irradiation with Kr^15+^ ions. At the same time, an increase in the radiation dose leads to almost the same difference between the types of radiation for doses of 10^14^–5 × 10^14^ ion/cm^2^, which makes it possible to estimate the contribution to the decrease in intensity depending on the type of ion, which is 8–9%. An increase in the irradiation dose to 10^15^ ion/cm^2^ leads to an increase in the difference to 15%, which indicates a greater number of structural deformations upon irradiation with Xe^22+^ ions in the case when the overlapping regions become larger.

The shift of the luminescence peak maxima for irradiated samples can also be caused by the appearance of the electron capture effect, the appearance of additional electron traps and a change in the electron density in ceramics as a result of irradiation. In the case of irradiation with heavy ions with an energy of more than 100 MeV, the main contribution to radiation defects is made by interactions with electron shells, due to the knocking out of electrons from the shells, and forming cascades of secondary electrons. In the case of high irradiation doses, when the probability of overlapping defect regions is high, the formed cascades of secondary electrons can make a significant contribution to the change in the electron density of ceramics, even taking into account the partial annihilation and relaxation of defects, as well as a small percentage of point defect survival during irradiation.

The determination of the effect of irradiation on the depth of occurrence of electron traps arising during irradiation was carried out using the technique of measuring TSL curves at different heating rates and plotting of the dependence of ln*(T_m_*^2^*/β)* to *1/kT_m_*, where *T_m_* is the temperature of the maximum of the TSL curves, *β* is the rate heating and *k* is the Boltzmann constant. According to Equation (1) and plotting the dependences, the values of the thermal depth of occurrence of electron traps (*E_T_*) were determined:(1)$$\ln\left(\frac{T_{m}^{2}}{\beta }\right)=\frac{E_{T}}{kT_{m}}+\ln\left(\frac{E_{T}}{sk}\right)$$
where *s* is the frequency factor. The results are shown in Figure 3.

For the initial sample of ceramics, the depth of occurrence of electron traps is 0.189 eV, which is in good agreement with the literature data for polycrystalline samples of AlN ceramics [33]. An increase in the irradiation dose for Kr^15+^ ions leads to a slight increase in the depth of occurrence, while for samples irradiated with Xe^22+^ ions, there is a sharp increase in the depth of occurrence from 5 to 20% depending on the irradiation dose. Such a large difference in the depth change of electron traps is primarily associated with a change in the electron density in the dielectric during the passage of heavy ions in the material, as well as structural distortions and the formation of vacancy defects. To assess the change in electron density, UV-Vis transmission spectra were measured, which reflect the change in the state of electrons based on a change in the position of the fundamental absorption edge. In the case of irradiation with heavy ions with an energy of more than 100 MeV, the main contribution to the energy loss is made by collisions with electron shells and, consequently, the subsequent change in the electronic configuration along the trajectory of the ions. At the same time, because of an increase in the radiation dose, which leads to an increase in the probability of overlapping the damaged areas, as well as in the ion energy, which is accompanied by large energy losses, the change in the electron density will be more pronounced.

Figure 4 shows the UV-Vis transmission spectra of the studied ceramics before and after irradiation with Kr^15+^ and Xe^22+^ ions.

For irradiated samples, the changes in the transmission spectra are associated with both a decrease in the spectral intensity and a shift of the fundamental absorption edge to the region of short wavelengths. A decrease in the intensity of the transmission spectra indicates a deterioration in the transmittance of ceramics, which is due to an increase in the dislocation and point defects in the structure of ceramics, as a result of defect formation during irradiation. The shift of the spectra leads to an increase in the band gap, which indicates a change in the electron density of the irradiated ceramics. Figure 5 shows the Tauc plots, which reflect the change in the band gap value as a result of irradiation. For the initial samples, the band gap is 3.98 eV. As can be seen from the presented data, the largest change in the band gap (*E*_g_) is observed for ceramics irradiated with Xe^22+^ ions, for which an increase in *E*_g_ varies from 7 to 20%, depending on the fluence. The change in the value of *E*_g_ for the samples irradiated with Kr^15+^ ions, in turn, varies from 1 to 10%. According to the data obtained, such a large difference in the change in the band gap is associated primarily with large radiation damage caused by irradiation with Xe^22+^ ions, which can result in the formation of disordering regions or highly defective regions in the structure of ceramics [34].

Figure 6 and Figure 7 show graphs of the refractive index change, as well as the Reflection loss *R*_L_ and optical transmission *T* values, the dynamics of which reflects the optical properties of the ceramics exposed to irradiation.

A change in the refractive index is known to be associated with a change in the defect structure of a material, its deformation as a result of external influences, or a change in density. In the case of ceramics, the change in the refractive index value, as well as reflection loss, *R*_L_, and optical transmission, *T*, is due to deformation processes of the crystal structure and distortion of the crystal lattice as a result of ionization losses of incident ions.

The change in the optical properties of AlN ceramics is primarily associated with a change in the defect density, as well as the processes of the recombination of oxygen O_N_ impurities and V_Al_ vacancies. The presence of oxygen vacancies in the structure of ceramics is due to the presence of the stabilizing additive Al_2_O_3_.

Figure 8 shows the results of changes in the UV-Vis absorption spectra of ceramics before and after irradiation. All the observed spectra are characterized by the presence of three spectral peaks with maxima at 3.4–3.5 eV, 4.5 eV and 6.2 eV, which correspond to defects in the ceramics structure associated with the formation of oxygen O_N_ impurities and V_Al_ vacancies, as well as electron traps [35]. For irradiated samples, an increase in these peaks is observed, which confirms an increase in the concentration of defects, complex defects of the O_N_-V_Al_ type and electron traps. Change in the peak intensity at 3.4–3.5 eV, 4.5 eV and 6.2 eV (Berzina et al. [21] and Slack et al. [36]) are associated primarily with a change in the concentration of defects and complexes of the O_N_-V_Al_ type in the structure above the valence band, as well as a change in the depth of electron traps as a result of external influences.

After analyzing the obtained spectra by the method of semiquantitative analysis of the peak areas characteristic of the complexes of defects, it was found that an increase in the irradiation dose leads to an increase in the concentration of O_N_-V_Al_ type defects at the maximum irradiation fluence by a factor of 5.9 for Kr^15+^ ions and 6.7 for Xe^22+^ ions.

Under irradiation with high-energy heavy ions, as mentioned earlier, the main contribution to the mechanisms of defect formation is made by interactions with electron shells, by transferring high energy to electrons, thereby knocking them out of their orbits. In the case of dielectrics, at high radiation doses, an electron density gradient may appear near the trajectory of ion movement in the material. In this case, the effect of overlapping ion trajectories can significantly change this gradient, and also form additional vacancy defects. Additionally, O_N_ impurity defects, which have a lower binding energy, can migrate over the structure due to the transferred energy of incident ions and form defect complexes. An increase in the energy of the incident ion, as well as in its mass and charge, leads to large structural and optical changes, which are caused by large energy losses, as well as a greater penetrating ability of ions, which is clearly reflected in the optical spectra. The shift of the fundamental absorption edge and a decrease in the transmission intensity indicate a change not only in the concentration of defects in the structure, which can have a negative effect on the optical properties, but also reflect a change in the electronic structure of dielectric ceramics.

Figure 9 shows the results of the change in the value of thermal conductivity, as well as the coefficient of efficiency of conservation of thermal conductivity, depending on the irradiation conditions. These characteristics reflect the heat-conducting properties and also show the resistance of the heat-insulating properties of ceramics to radiation.

As can be seen from the presented data, the greatest changes in thermal conductivity are observed at the large irradiation fluences of 5 × 10^14^–1 × 10^15^ ion/cm^2^. In this case, for the samples irradiated with Xe^22+^ ions, the largest decrease in thermal conductivity was 19%, which is close to the critical values of the heat-conducting properties of the insulating materials. At the same time, for ceramics irradiated with Kr^15+^ ions, the maximum decrease was no more than 10%.

Figure 10 shows comparative data on changes in the position, shape and intensity of the three main reflexes, which most fully characterize the structural changes as a result of irradiation. According to the data obtained, it can be seen that, as a result of irradiation, the main changes are associated with changes in the intensities, shape and position of diffraction maxima, which indicates a change in the concentration of point defects, dislocations and distortions of the crystal structure during irradiation. In this case, the appearance of any new diffraction reflections is not observed, which indicates the absence of phase transformation processes as a result of irradiation. According to earlier studies [13,26,27,28,29,30,31], these ceramics are characterized by a hexagonal type of the crystal structure with three main textural directions, (100), (002) and (101), the change in the positions of which characterizes the main changes associated with the crystal lattice and its parameters *a* and *c*. In this case, a change in the position of the diffraction maxima (100) and (101) is characteristic of the deformation of the crystal lattice parameter a, and a change in the position of the (002) maximum is characteristic of the deformation of the lattice along the *c* axis. A change in the intensity and shape of the diffraction line indicates a change in distortions and stresses in the crystal lattice, as well as the influence of the size factor associated with the size of crystallites.

According to the presented data, it can be clearly seen that an increase in the radiation dose leads to a decrease in the intensity of the diffraction lines and their asymmetry, which indicates a change in the size of crystallites, their fragmentation or reorientation as well as distortion and deformation of the structure. In this case, the shift of the diffraction maxima to the region of small angles 2θ indicates an increase in the interplanar distances, as well as the deformation of atomic planes as a result of the appearance of primarily displaced atoms and defect regions in the structure. A decrease in the intensity and a change in the half-width of the diffraction lines (FWHM) indicate a change in the crystallite size, a decrease in which leads to an increase in the dislocation density in the irradiated layer, as well as the appearance of vacancy and point defects. In this case, in contrast to ceramics irradiated with Kr^15+^ ions, irradiation with Xe^22+^ ions leads to large changes in the diffraction patterns, which indicates large structural distortions and deformations. Figure 11 shows the results of assessing the change in dislocation density as a result of irradiation, as well as distortions of interplanar spacings as a result of displacement of diffraction maxima, which also characterize the deformation of the crystal lattice.

As can be seen from the presented data, an increase in the irradiation dose, as well as a change in the type of ion upon irradiation, leads to an increase in the value of crystal lattice distortions along both axes, a and c, as well as to an increase in the dislocation density in the structure of the irradiated layer. In this case, in the case of irradiation with Xe^22+^ ions, the change in the dislocation density is exponential, depending on the irradiation dose; at the maximum irradiation dose, the dislocation density increases by a factor of 3, while when irradiated with Kr^15+^ ions, it is only 1.7 times as compared to the initial dislocation density. Such a significant difference in the dislocation density with a change in the type of ion upon irradiation can be associated with the large structural distortions during irradiation with Xe^22+^ ions, as well as the resulting regions of disordering and amorphous inclusions, which are evidenced by the broadening of diffraction peaks and their asymmetric shape. Additionally, in the case of irradiation with Xe^22+^ ions at high irradiation fluences, a significant crushing of crystallites and a decrease in their size from 170–175 nm for the initial sample to 102–105 nm for a sample irradiated with a fluence of 10^15^ ion/cm^2^ were observed.

In the case of irradiation with heavy high-energy ions, the distribution of defects in the entire irradiated volume is uniform. At the same time, according to previous studies, it was found that the subsurface layer undergoes the greatest change, in which craters and hillocks can form, the formation of which is associated both with the processes of partial surface sputtering, as well as partial extrusion of the defective volume onto the surface. Previously, in [13], we demonstrated that upon irradiation with heavy ions, defective inclusions are distributed throughout the entire volume, with the formation of fused grains and regions of disorder.

The presented results of changes in the structural characteristics are in good agreement with changes in the optical and heat-conducting properties of ceramics irradiated with heavy Kr^15+^ and Xe^22+^ ions, and the totality of all the results obtained will further make it possible to contribute to the development of the theory of radiation damage of nitride ceramics and to determine the range of their applicability.

## 4. Conclusions

This work presents the results of a study of changes in the optical and heat-conducting properties of AlN nitride ceramics irradiated with Kr^15+^ and Xe^22+^ heavy ions with energies of 140 MeV and 220 MeV and fluences of 10^14^–10^15^ ion/cm^2^. During the study, dose dependences of changes in the optical properties of ceramics, as well as the influence of the type of ions on the degree of radiation damage and deterioration of optical characteristics were established. It was found that an increase in the irradiation dose for Kr^15+^ ions leads to an insignificant increase in the depth of occurrence of electron traps, while for samples irradiated with Xe^22+^ ions, a sharp increase in the depth of occurrence from 5 to 20% is observed, depending on the radiation dose. It was found that the shift of the fundamental absorption edge and a decrease in the transmission intensity indicate a change not only in the concentration of defects in the structure, which can have a negative effect on the optical properties, but also reflect a change in the electronic structure of dielectric ceramics. Using the X-ray diffraction method, it was found that in the case of irradiation with Xe^22+^ ions, the change in the dislocation density is exponential depending on the irradiation dose, which indicates the processes of recrystallization and fragmentation of grains at high irradiation doses.

Further research will be aimed at studying the possibilities of increasing the radiation resistance of ceramics to various types of ionizing radiation.

## Figures and Tables

**Figure 1 nanomaterials-10-02375-f001:**
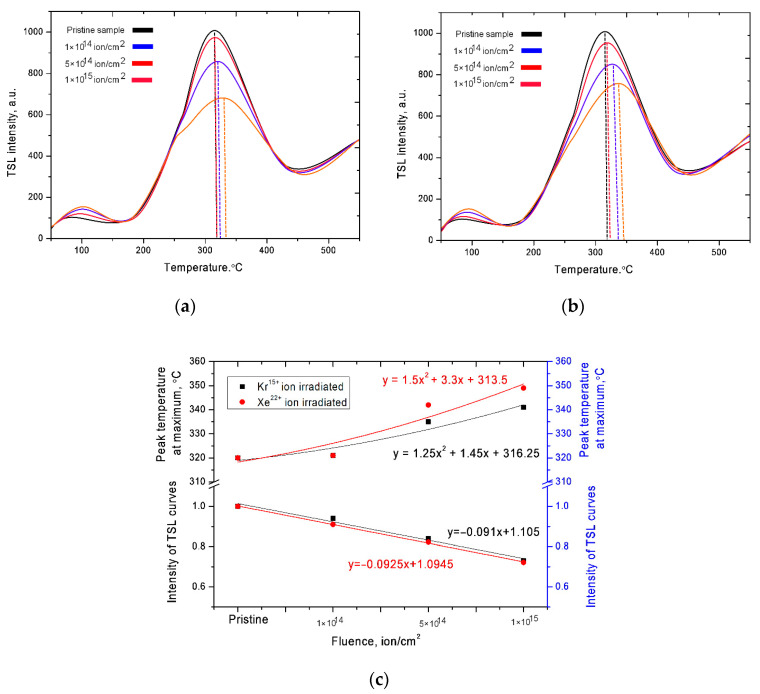
Thermally stimulated luminescence (TSL) curves of the studied ceramics depending on the heating temperature: (**a**) Kr^15+^ ion irradiated dose; (**b**) Xe^22+^ ion irradiated dose; (**c**) Graph of changes in the TSL curves’ intensity and maximum temperature depending on the irradiation fluence.

**Figure 2 nanomaterials-10-02375-f002:**
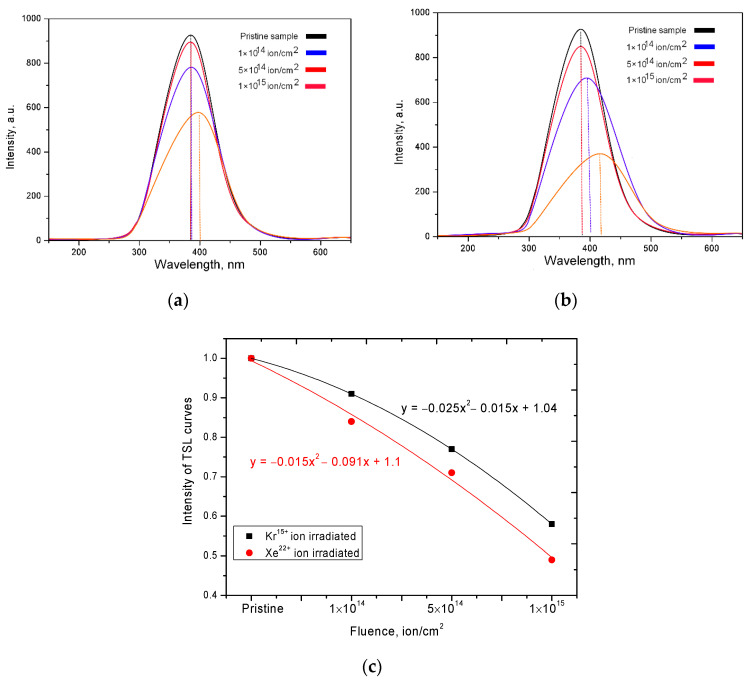
TSL curves versus wavelength: (**a**) Kr^15+^ ion irradiated dose; (**b**) Xe^22+^ ion irradiated dose; (**c**) Graph of the TSL curves’ intensity change depending on irradiation fluence.

**Figure 3 nanomaterials-10-02375-f003:**
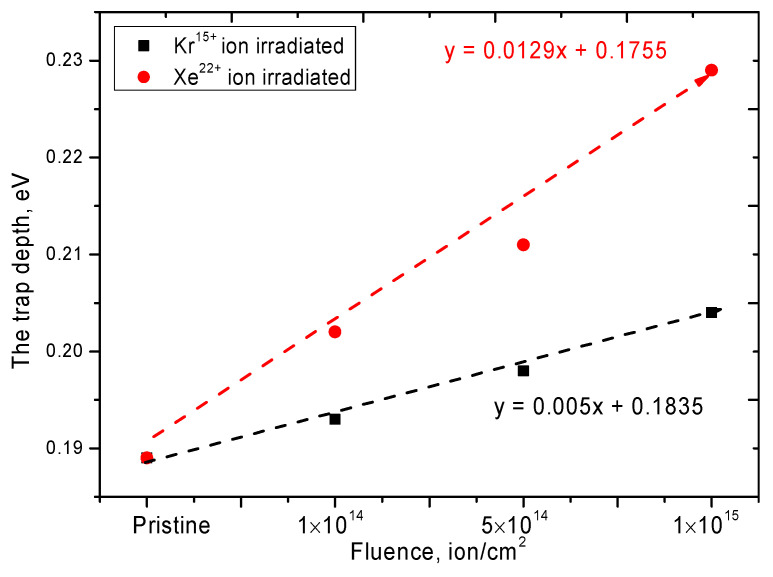
Graph of the electron trap depth change versus radiation dose.

**Figure 4 nanomaterials-10-02375-f004:**
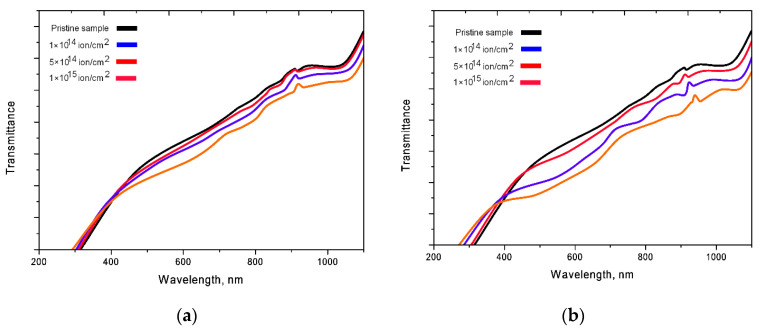
UV–visible transmittance of AlN ceramic: (**a**) Kr^15+^ ion irradiated dose; (**b**) Xe^22+^ ion irradiated dose.

**Figure 5 nanomaterials-10-02375-f005:**
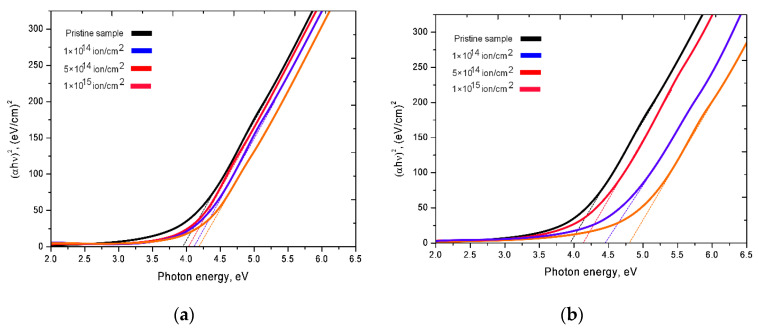
Tauc’s plot obtained from UV-vis transmittance spectra for evaluating the optical band gap of AlN: (**a**) Kr^15+^ ion irradiated dose; (**b**) Xe^22+^ ion irradiated dose.

**Figure 6 nanomaterials-10-02375-f006:**
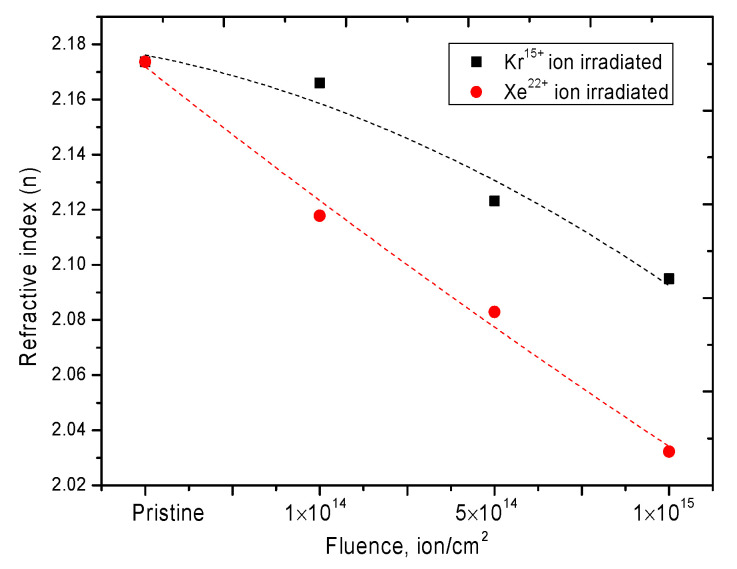
Graph of the refractive index changes depending on the radiation dose.

**Figure 7 nanomaterials-10-02375-f007:**
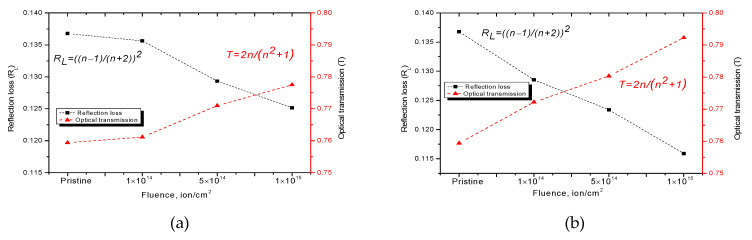
Reflection loss *R*_L_ and optical transmission *T* for: (**a**) the Kr^15+^ ion irradiated dose; (**b**) the Xe^22+^ ion irradiated dose.

**Figure 8 nanomaterials-10-02375-f008:**
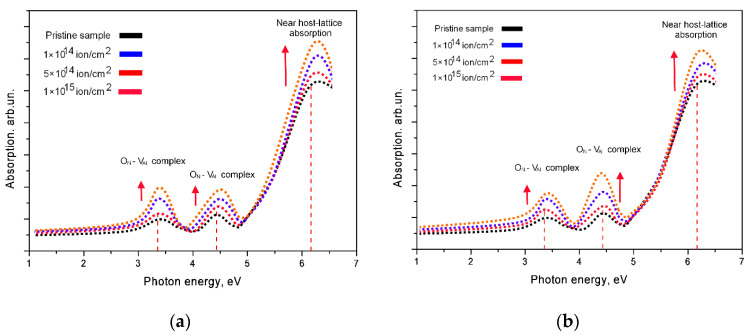
UV-Vis absorption spectra of ceramics before and after irradiation: (**a**) Kr^15+^ ion irradiated dose; (**b**) Xe^22+^ ion irradiated dose; (**c**) Semi-quantitative analysis of the concentration of complex defects depending on the type of irradiation.

**Figure 9 nanomaterials-10-02375-f009:**
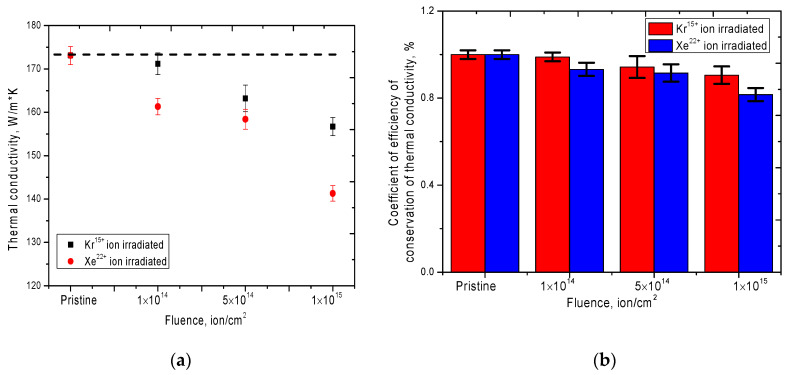
(**a**) Graph of the dependence of changes in the thermal conductivity of ceramics after irradiation; (**b**) Diagram of changes in the coefficient of efficiency of thermal conductivity conservation depending on the irradiation conditions.

**Figure 10 nanomaterials-10-02375-f010:**
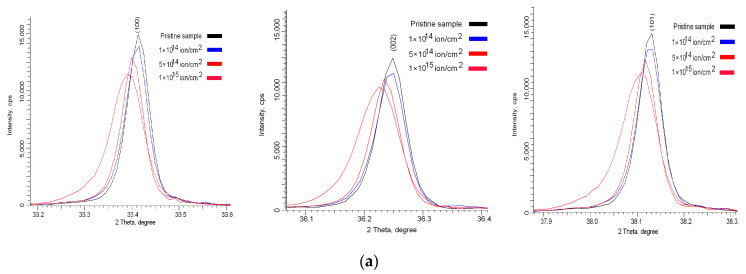
Detailed change of diffraction maxima (100), (002) and (101): (**a**) Kr^15+^ ion irradiated dose; (**b**) Xe^22+^ ion irradiated dose.

**Figure 11 nanomaterials-10-02375-f011:**
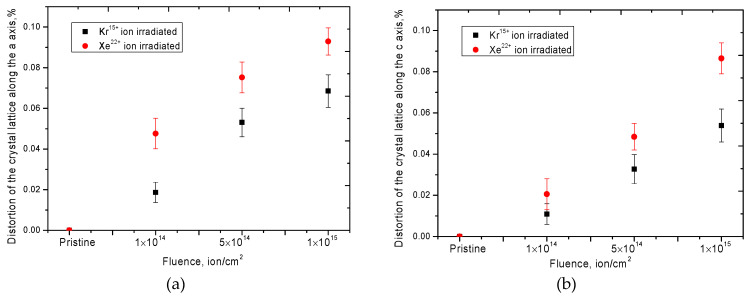
(**a**) Graph of crystal lattice distortion along the *a* axis; (**b**) Graph of crystal lattice distortion along the *c* axis; (**c**) The graph of the change of the dislocation density depending on the radiation dose.

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
