# Peer review of "Study of Changes in Optical and Heat-Conducting Properties of AlN Ceramics under Irradiation with Kr15+ and Xe22+ Heavy Ions"

_nanomaterials, 2020, doi:10.3390/nano10122375_

Round 1
Reviewer 1 Report
The authors report the change in the optical properties of AlN ceramics induced by irradiation with heavy ions. From the viewpoint of applications of AlN ceramics, the information presented in this paper is interesting to readers of Nanomaterials. The change in the optical properties have sufficiently been analyzed from the viewpoint of optical absorption/transmission spectra and thermally stimulated luminescence (TSL) properties. The paper can be published after minor modifications. I hope the author clarify the following points:
- In the 3rd paragraph of Introduction, "increasing intensity of optical and luminescent properties" is unclear to me. Do you mean an enhanced absorbance (for optical absorption) or intensity (for PL or TL)?
- I found some papers on ion irradiation effects (proton, carbon, Fe, etc.) of AlN. The authors should refer to the papers to clearly present the novelty of this paper in Introduction.
- The supplier of the AlN ceramics should be provided. AlN ceramics sometimes have different optical (in particular, PL) properties possibly owing to the difference in the fabrication procedure (sintering temperature or atmosphere) and stabilizer included in the ceramics.
- In the 2nd paragraph of Experimental part, "For the selected radiation doses, the probability of overlap varies from 100 to 1000 times overlap" needs some additional explanation: how did you estimate the defect region of the experimental condition in this study?
- Was TSL observed just after irradiation with heavy ions or after annealing? On this context, I cannot understand "after initialization with X-ray radiation" in the 3rd paragraph of Experimental part.
- In the caption of Figure 1, "TSL spectra" should be revised to "TSL curves". The same revision should be applied to some places in the main text at which you intend to call TSL (glow) curves.
- I cannot agree "the TSL spectrum is characterized by the presence of a small peak in the region of 80-110°C, which is characteristic of rearrangement defects", because "rearrangement of defects" usually means a displacement of atoms or ions around the defects. In this case, the low-temperature glow peak corresponds to the release of shallow trapped electrons from defects.
- How were the TSL spectra obtained (device used in the measurements, recorded temperature range, etc.)?
- The change in the TSL spectra should be discussed on the basis of possible origin of the TSL emission band.
- How do you explain the gradual increase in the electron trap depth with the dose? The trap depths are different for different trapping sites. If the irradiation generated defects, a glow peak having the trap depth corresponding to the generated defects would be observed. The gradual increase in the trap depth may be caused by the change in the environment of the electron trapping sites.
- The increase in the band-gap energy with the dose needs further discussion. The authors attributed the increase in the band-gap energy to disordering caused by the heavy ions. In this case, analysis of the lattice structure after the irradiation (e.g. XRD measurements) would be helpful for discussion. As far as I know, the absorption tail shifts toward the longer wavelength after amorphization of many insulators.
Author Response
|
1. The authors thank the reviewer for this comment. The text of the article has been corrected as follows: Thus, for example, a series of works by Trinkler, L., et al. is devoted to the study of changes in optical and luminescent properties by modification with UV and gamma radiation [22-25]. The authors studied in detail the mechanisms of luminescence change in AlN, which are associated with the recombination processes and changes in the concentration of defects in the crystal lattice.
|
||||||||||||||||||||||||
|
2. The authors thank the reviewer for the interest shown in this topic of the study. The introduction added a description of the relevance of the study, referring to previous studies. However, while the optical and luminescent properties of AlN ceramics have been studied, there are still many questions regarding the effects of ion bombardment defect formation processes on the properties of ceramics. The aim of this work is to study the effect of irradiation with heavy ions Kr15+ and Xe22+ with energies of 140 MeV and 220 MeV and fluences of 1014 – 1015 ion/cm2 on the optical and heat-conducting properties of aluminum nitride ceramics. This work is part of the research cycle of our scientific group devoted to the study of the physical and chemical, structural, optical and mechanical properties of ceramics exposed to ionizing radiation. Earlier, our research group studied in detail the changes in the structural and mechanical properties of AlN ceramics exposed to irradiation of heavy Fe7+ ions with energies up to 100 MeV [26,27]. The mechanisms of helium swelling and processes of hydrogenation of the near-surface layer exposed to high-dose irradiation (1016-5x1017 ion/cm2) have been studied [28-30]. Studies have been carried out to investigate the effect of the formation of impurity inclusions as a result of ion implantation on the optical and structural properties of ceramics irradiated with low-energy С2+ ions [31]. During these studies, the mechanisms of defect formation in the near-surface layer were established, as well as the effect of defects on the change in the properties of ceramics. The results obtained have shown a high resistance of these ceramics to radiation exposure, as well as to processes of degradation of the near-surface layer as a result of the accumulation of radiation-stimulated defects, which opens up prospects for using these materials in nuclear power engineering as structural materials. A distinctive feature of this research is a detailed study of the effect of defect formation under irradiation with heavy ions with energies of 140-220 MeV, comparable to fragments of uranium fission on the optical and luminescent properties of ceramics. As is known, in dielectrics, a change in the optical properties is directly related to a change in the electron density, as well as structural and impurity defects. At the same time, in the case of radiation damage arising from irradiation with heavy ions with an energy of more than 1.0 MeV/nucleon, the greatest contribution to the creation of defects is made by elastic collisions of ions with electron shells.
|
||||||||||||||||||||||||
|
3. Commercial polycrystalline aluminum nitride ceramics stabilized with aluminum oxide (not more than 4 at%), which are widely used as an insulating material, a base for microelectronic devices, and also as structural materials for nuclear power, were chosen as the samples under study. Samples were purchased from Shenzhen JRFT, China. According to the manufacturer's data, the ceramics were obtained by sintering Al2O3 powders in a nitrogen (N2) atmosphere at a temperature of 1600-1800°C and a pressure of P = 760 mmHg. |
||||||||||||||||||||||||
|
According to the literature [10,32], in the case of irradiation with heavy ions with an energy of more than 100 MeV, when a heavy ion passes through the target material, the radius of the defect region, which in some cases is called a latent track, is estimated to be from 3 to 10-20 nm. This region is a defect structure arising from the interaction of an incident ion with the electronic and nuclear subsystem of the target. An increase in the irradiation fluence above 1013 ion/cm2 leads to overlapping of these regions [10,32]. |
||||||||||||||||||||||||
|
The thermally stimulated luminescence (TSL) curves of ceramics before and after irradiation were measured in the temperature range from 50 to 550°C. For the original samples, luminescence was observed after initialization with X-ray radiation (55 kV, 10mA) for 10 minutes. For irradiated samples, luminescence was observed after irradiation. The luminescence was recorded using a Hamamatsu R6357 photomultiplier tube (Hamamatsu Photonics K.K., Shimokanzo, Japan) with a sensitivity in the range of 200-900 nm. |
||||||||||||||||||||||||
|
6. The authors thank the reviewer for this comment. The text of the article has been corrected and replaced with expressions. |
||||||||||||||||||||||||
|
7. The authors thank the reviewer for this comment. The text of the article has been corrected and replaced with expressions. For the initial samples, the TSL curve is characterized by the presence of a small peak in the region of 80-110°C, which corresponds to the release of trapped electrons from the traps, and a large peak in the region of 280-380°C with a maximum at 320°C. |
||||||||||||||||||||||||
|
The thermally stimulated luminescence (TSL) curves of ceramics before and after irradiation were measured in the temperature range from 50 to 550°C. For the original samples, luminescence was observed after initialization with X-ray radiation (55 kV, 10mA) for 10 minutes. For irradiated samples, luminescence was observed after irradiation. The luminescence was recorded using a Hamamatsu R6357 photomultiplier tube (Hamamatsu Photonics K.K., Shimokanzo, Japan) with a sensitivity in the range of 200-900 nm. |
||||||||||||||||||||||||
|
9. The authors thank the reviewer for this comment. The text of the article has been corrected and replaced with expressions. For the initial samples, the TSL curve is characterized by the presence of a small peak in the region of 80-110°C, which corresponds to the release of trapped electrons from the traps, and a large peak in the region of 280-380°C with a maximum at 320°C. An increase in the intensity of the TSL curve above a temperature of 500 ° C is due to the thermal radiation of the samples during heating. For irradiated samples at irradiation fluences of 1x1014 ion/cm2, a decrease in intensity and an insignificant shift of the maximum toward higher temperatures are observed. A further increase in the radiation dose leads to an increase in the trends of displacement of the maxima and a decrease in their intensities (see Figure 1c). Also, for irradiation doses of 5х1014-1х1015 ion/cm2, there is a slight increase in the peak intensity in the range of 80-110°C, which indicates a change in the defect structure in ceramics at high irradiation doses. In turn, the shift of the maximum to the region of high temperatures is also due to an increase in the defect structure of ceramics as a result of irradiation, as well as a change in the depth of occurrence of electron traps and their activation energy in irradiated ceramics. In the case of irradiation with heavy ions, when they pass through the material in small areas along the trajectory of the ions, a large gradient of the electron density arises due to the large energy losses of the ions. In this case, the overlap of these regions leads to an increase in the concentration of defects associated with the creation of electron-hole pairs, regions with local electronic excitation, breaking of chemical and crystal bonds, as well as atomic diffusion in the damaged layer. The appearance of defect regions leads to a change in the activation energy and the depth of occurrence of electron traps, which results into an increase in the activation temperature, as well as a decrease in the area under the curve (light sum), which is associated with a change in the band gap of deep electron traps. Also, in the case of large energy losses of incident ions and, consequently, large energy transfers to the electronic and nuclear subsystems, it can lead to the formation of not only cascades of elastic collisions, but also the formation of amorphous regions or regions of disorder, which also have a negative effect on optical and luminescent properties of ceramics. |
||||||||||||||||||||||||
|
10. For the initial sample of ceramics, the depth of occurrence of electron traps is 0.189 eV, which is in good agreement with the literature data for polycrystalline samples of AlN ceramics [33]. An increase in the irradiation dose for Kr15+ ions leads to a slight increase in the depth of occurrence, while for samples irradiated with Xe22+ ions there is a sharp increase in the depth of occurrence from 5 to 20 % depending on the irradiation dose. Such a large difference in the depth change of electron traps is primarily associated with a change in the electron density in the dielectric during the passage of heavy ions in the material, as well as structural distortions and the formation of vacancy defects. To assess the change in electron density, UV-Vis transmission spectra were measured, which reflect the change in the state of electrons based on a change in the position of the fundamental absorption edge. In the case of irradiation with heavy ions with an energy of more than 100 MeV, the main contribution to the energy loss is made by collisions with electron shells and, consequently, the subsequent change in the electronic configuration along the trajectory of the ions. At the same time, an increase in the radiation dose, which leads to an increase in the probability of overlapping the damaged areas, as well as the ion energy, which is accompanied by large energy losses, the change in the electron density will be more pronounced.
|
||||||||||||||||||||||||
|
11. Figure 10 shows X-ray diffraction patterns of the samples under study before and after irradiation. According to the data obtained, it can be seen that, as a result of irradiation, the main changes are associated with changes in the intensities, shape and position of diffraction maxima, which indicates a change in the concentration of point defects, dislocations, and distortions of the crystal structure during irradiation. In this case, the appearance of any new diffraction reflections is not observed, which indicates the absence of phase transformation processes as a result of irradiation. According to earlier studies [13,26-31], it is known that these ceramics are characterized by a hexagonal type of crystal structure with three main textural directions (100), (002) and (101), the change in the positions of which characterizes the main changes associated with the crystal lattice and its parameters a and c. In this case, a change in the position of the diffraction maxima (100) and (101) is characteristic of the deformation of the crystal lattice parameter a, and a change in the position of the (002) maximum is characteristic of the deformation of the lattice along the c axis. A change in the intensity and shape of the diffraction line indicates a change in distortions and stresses in the crystal lattice, as well as the influence of the size factor associated with the size of crystallites.
Figure 10. X-ray diffraction patterns of the studied ceramics: a) Kr15+ ion irradiated; b) Xe22+ ion irradiated. Figure 11 shows comparative data on changes in the position, shape and intensity of the three main reflexes, which most fully characterize the structural changes as a result of irradiation.
Figure 11. Detailed change of diffraction maxima (100), (002) and (101): a) Kr15+ ion irradiated; b) Xe22+ ion irradiated. According to the presented data, it is clearly seen that an increase in the radiation dose leads to a decrease in the intensity of diffraction lines and their asymmetry, which indicates a change in the size of crystallites, their fragmentation or reorientation, as well as distortion and deformation of the structure. In this case, the shift of the diffraction maxima to the region of small angles 2θ indicates an increase in the interplanar distances, as well as the deformation of atomic planes as a result of the appearance of primarily displaced atoms and defect regions in the structure. A decrease in the intensity and a change in the half-width of the diffraction lines (FWHM) indicate a change in the crystallite size, a decrease in which leads to an increase in the dislocation density in the irradiated layer, as well as the appearance of vacancy and point defects. In this case, in contrast to ceramics irradiated with Kr15+ ions, irradiation with Xe22+ ions leads to large changes in the diffraction patterns, which indicates large structural distortions and deformations. Figure 12 shows the results of assessing the change in dislocation density as a result of irradiation, as well as distortions of interplanar spacings as a result of displacement of diffraction maxima, which also characterize the deformation of the crystal lattice.
Figure 12. a) Graph of crystal lattice distortion along the a axis; b) Graph of crystal lattice distortion along the c axis; c) The graph of the change in dislocation density depending on the radiation dose
As can be seen from the presented data, an increase in the irradiation dose, as well as a change in the type of ion upon irradiation, leads to an increase in the value of crystal lattice distortions along both axes a and c, as well as to an increase in the dislocation density in the structure of the irradiated layer. In this case, in the case of irradiation with Xe22+ ions, the change in the dislocation density is exponential depending on the irradiation dose; at the maximum irradiation dose, the dislocation density increases by a factor of 3, while when irradiated with Kr15+ ions, it is only 1.7 times as compared to the initial dislocation density. Such a significant difference in the dislocation density with a change in the type of ion upon irradiation can be associated with large structural distortions during irradiation with Xe22+ ions, as well as the resulting regions of disordering and amorphous inclusions, which are evidenced by the broadening of diffraction peaks and their asymmetric shape. Also, in the case of irradiation with Xe22+ ions at high irradiation fluences, a significant crushing of crystallites and a decrease in their size from 170-175 nm for the initial sample to 102-105 nm for a sample irradiated with a fluence of 1015 ion/cm2 were observed. The presented results of changes in the structural characteristics are in good agreement with changes in the optical and heat-conducting properties of ceramics irradiated with heavy Kr15+ and Xe22+ ions, and the totality of all the results obtained will further make it possible to contribute to the development of the theory of radiation damage of nitride ceramics and to determine the range of their applicability.
|
||||||||||||||||||||||||

Reviewer 2 Report
See attached document

Author Response
|
1. The authors thank the reviewer for this comment. Corrections are made to the text of the article. |
||||||||||||||||||||||||||||||||
|
2. The authors thank the reviewer for this comment. Corrections are made to the text of the article. |
||||||||||||||||||||||||||||||||
|
3. The authors thank the reviewer for this comment. Corrections are made to the text of the article.
The optical and luminescent properties of nitride ceramics deserve special attention when considering nitride ceramics and their practical application [16–18]. Interest in them is due to the possibility to use them in the field of above mentioned applications. |
||||||||||||||||||||||||||||||||
|
4. Commercial polycrystalline AlN ceramics stabilized with aluminum oxide (not more than 4 at%), which are widely used as an insulating material. Samples were purchased from Shenzhen JRFT, China. According to the manufacturer's data, the ceramics were obtained by sintering Al2O3 powders in a nitrogen (N2) atmosphere at a temperature of 1600-1800°C and a pressure of P = 760 mmHg. The thickness of the samples was 20 μm, the density was 3.26 g/cm3, the structure type was hexagonal, the symmetry type was P63mc(186); according to X-ray diffraction analysis, the grain size was 170-175 nm, and this size is in good agreement with the observed sizes using a scanning electron microscopy. Before irradiation, the samples were subjected to mechanical polishing of the surface in order to obtain a roughness of no more than 5-10 nm. The choice of the sample thickness of 20 μm is due to the path length of ions in the ceramic in order to simulate defects in the entire irradiated volume. |
||||||||||||||||||||||||||||||||
|
5. The authors thank the reviewer for this comment. Corrections are made to the text of the article.
|
||||||||||||||||||||||||||||||||
|
6. The authors thank the reviewer for this comment. Corrections are made to the text of the article.
|
||||||||||||||||||||||||||||||||
|
7. The authors thank the reviewer for this comment. Corrections are made to the text of the article.
Figure 1 shows the TSL curves of ceramics before and after irradiation with heavy ions with different fluences.
Figure 1. TSL curves of the studied ceramics depending on the heating temperature: a) Kr15+ ion irradiated; b) Xe22+ ion irradiated; c) Graph of changes in TSL curves intensity and maximum temperature depending on irradiation fluence For the initial samples, the TSL curve is characterized by the presence of a small peak in the region of 80-110°C, which corresponds to the release of trapped electrons from the traps, and a large peak in the region of 280-380°C with a maximum at 320°C. An increase in the intensity of the TSL curve above a temperature of 500 ° C is due to the thermal radiation of the samples during heating. For irradiated samples at irradiation fluences of 1x1014 ion/cm2, a decrease in intensity and an insignificant shift of the maximum toward higher temperatures are observed. A further increase in the radiation dose leads to an increase in the trends of displacement of the maxima and a decrease in their intensities (see Figure 1c). Also, for irradiation doses of 5х1014-1х1015 ion/cm2, there is a slight increase in the peak intensity in the range of 80-110°C, which indicates a change in the defect structure in ceramics at high irradiation doses. In turn, the shift of the maximum to the region of high temperatures is also due to an increase in the defect structure of ceramics as a result of irradiation, as well as a change in the depth of occurrence of electron traps and their activation energy in irradiated ceramics. In the case of irradiation with heavy ions, when they pass through the material in small areas along the trajectory of the ions, a large gradient of the electron density arises due to the large energy losses of the ions. In this case, the overlap of these regions leads to an increase in the concentration of defects associated with the creation of electron-hole pairs, regions with local electronic excitation, breaking of chemical and crystal bonds, as well as atomic diffusion in the damaged layer. The appearance of defect regions leads to a change in the activation energy and the depth of occurrence of electron traps, which results into an increase in the activation temperature, as well as a decrease in the area under the curve (light sum), which is associated with a change in the band gap of deep electron traps. Also, in the case of large energy losses of incident ions and, consequently, large energy transfers to the electronic and nuclear subsystems, it can lead to the formation of not only cascades of elastic collisions, but also the formation of amorphous regions or regions of disorder, which also have a negative effect on optical and luminescent properties of ceramics.
Figure 2c shows a graph of the dependence of the change in the intensity of TSL curves on the radiation dose for various types of ions. As can be seen from this graph, in the case of irradiation with Xe22+ ions, the decrease in intensity is more pronounced than in the case of irradiation with Kr15+ ions. At the same time, an increase in the radiation dose leads to almost the same difference between the types of radiation for doses of 1014-5x1014 ion/cm2, which makes it possible to estimate the contribution to the decrease in intensity depending on the type of ion, which is 8-9 %. An increase in the irradiation dose to 1015 ion/cm2 leads to an increase in the difference to 15 %, which indicates a greater number of structural deformations upon irradiation with Xe22+ ions in the case when the overlapping regions become larger.
After analyzing the obtained spectra by the method of semiquantitative analysis of the peak areas characteristic of the complexes of defects, it was found that an increase in the irradiation dose leads to an increase in the concentration of ON-VAl type defects at the maximum irradiation fluence by a factor of 5.9 for Kr15+ ions and 6.7 times for Xe22+ ions. Figure 10 shows X-ray diffraction patterns of the samples under study before and after irradiation. According to the data obtained, it can be seen that, as a result of irradiation, the main changes are associated with changes in the intensities, shape and position of diffraction maxima, which indicates a change in the concentration of point defects, dislocations, and distortions of the crystal structure during irradiation. In this case, the appearance of any new diffraction reflections is not observed, which indicates the absence of phase transformation processes as a result of irradiation. According to earlier studies [13,26-31], it is known that these ceramics are characterized by a hexagonal type of crystal structure with three main textural directions (100), (002) and (101), the change in the positions of which characterizes the main changes associated with the crystal lattice and its parameters a and c. In this case, a change in the position of the diffraction maxima (100) and (101) is characteristic of the deformation of the crystal lattice parameter a, and a change in the position of the (002) maximum is characteristic of the deformation of the lattice along the c axis. A change in the intensity and shape of the diffraction line indicates a change in distortions and stresses in the crystal lattice, as well as the influence of the size factor associated with the size of crystallites.
Figure 10. X-ray diffraction patterns of the studied ceramics: a) Kr15+ ion irradiated; b) Xe22+ ion irradiated. Figure 11 shows comparative data on changes in the position, shape and intensity of the three main reflexes, which most fully characterize the structural changes as a result of irradiation.
Figure 11. Detailed change of diffraction maxima (100), (002) and (101): a) Kr15+ ion irradiated; b) Xe22+ ion irradiated. According to the presented data, it is clearly seen that an increase in the radiation dose leads to a decrease in the intensity of diffraction lines and their asymmetry, which indicates a change in the size of crystallites, their fragmentation or reorientation, as well as distortion and deformation of the structure. In this case, the shift of the diffraction maxima to the region of small angles 2θ indicates an increase in the interplanar distances, as well as the deformation of atomic planes as a result of the appearance of primarily displaced atoms and defect regions in the structure. A decrease in the intensity and a change in the half-width of the diffraction lines (FWHM) indicate a change in the crystallite size, a decrease in which leads to an increase in the dislocation density in the irradiated layer, as well as the appearance of vacancy and point defects. In this case, in contrast to ceramics irradiated with Kr15+ ions, irradiation with Xe22+ ions leads to large changes in the diffraction patterns, which indicates large structural distortions and deformations. Figure 12 shows the results of assessing the change in dislocation density as a result of irradiation, as well as distortions of interplanar spacings as a result of displacement of diffraction maxima, which also characterize the deformation of the crystal lattice.
Figure 12. a) Graph of crystal lattice distortion along the a axis; b) Graph of crystal lattice distortion along the c axis; c) The graph of the change in dislocation density depending on the radiation dose
As can be seen from the presented data, an increase in the irradiation dose, as well as a change in the type of ion upon irradiation, leads to an increase in the value of crystal lattice distortions along both axes a and c, as well as to an increase in the dislocation density in the structure of the irradiated layer. In this case, in the case of irradiation with Xe22+ ions, the change in the dislocation density is exponential depending on the irradiation dose; at the maximum irradiation dose, the dislocation density increases by a factor of 3, while when irradiated with Kr15+ ions, it is only 1.7 times as compared to the initial dislocation density. Such a significant difference in the dislocation density with a change in the type of ion upon irradiation can be associated with large structural distortions during irradiation with Xe22+ ions, as well as the resulting regions of disordering and amorphous inclusions, which are evidenced by the broadening of diffraction peaks and their asymmetric shape. Also, in the case of irradiation with Xe22+ ions at high irradiation fluences, a significant crushing of crystallites and a decrease in their size from 170-175 nm for the initial sample to 102-105 nm for a sample irradiated with a fluence of 1015 ion/cm2 were observed. In the case of irradiation with heavy high-energy ions, the distribution of defects in the entire irradiated volume is uniform. At the same time, according to previous studies, it was found that the subsurface layer undergoes the greatest change, in which craters and hillocks can form, the formation of which is associated both with the processes of partial surface sputtering, as well as partial extrusion of the defective volume onto the surface. In [13], we previously demonstrated that upon irradiation with heavy ions, defective inclusions are distributed throughout the entire volume, with the formation of fused grains and regions of disorder. The presented results of changes in the structural characteristics are in good agreement with changes in the optical and heat-conducting properties of ceramics irradiated with heavy Kr15+ and Xe22+ ions, and the totality of all the results obtained will further make it possible to contribute to the development of the theory of radiation damage of nitride ceramics and to determine the range of their applicability.
|
||||||||||||||||||||||||||||||||
|
8. Before irradiation, the samples were subjected to mechanical polishing of the surface in order to obtain a roughness of no more than 5-10 nm. The choice of the sample thickness of 20 μm is due to the path length of ions in the ceramic in order to simulate defects in the entire irradiated volume. |
||||||||||||||||||||||||||||||||
|
9. In the case of irradiation with heavy high-energy ions, the distribution of defects in the entire irradiated volume is uniform. At the same time, according to previous studies, it was found that the subsurface layer undergoes the greatest change, in which craters and hillocks can form, the formation of which is associated both with the processes of partial surface sputtering, as well as partial extrusion of the defective volume onto the surface. In [13], we previously demonstrated that upon irradiation with heavy ions, defective inclusions are distributed throughout the entire volume, with the formation of fused grains and regions of disorder. |
||||||||||||||||||||||||||||||||
|
10. The authors thank the reviewer for this comment. Corrections are made to the text of the article.
|
||||||||||||||||||||||||||||||||

Round 2
Reviewer 2 Report
Authors have made the appropriate modifications in the manuscript to improve its quality and it can be accepted in its present format. As comment, in Figure 10 we can not appreciate the effects of irradiation, and the detail of some diffraction peaks shown in Figure 11 is sufficient. They should remove Figure 10.
Author Response
The authors thank the referee for the work done. Corrections were made according to the comment of the reviewer.